# Recent Developments in Glioblastoma Therapy: Oncolytic Viruses and Emerging Future Strategies

**DOI:** 10.3390/v15020547

**Published:** 2023-02-16

**Authors:** Azzam Hamad, Gaukhar M. Yusubalieva, Vladimir P. Baklaushev, Peter M. Chumakov, Anastasiya V. Lipatova

**Affiliations:** 1Center for Precision Genome Editing and Genetic Technologies for Biomedicine, Engelhardt Institute of Molecular Biology, Russian Academy of Sciences, 119991 Moscow, Russia; 2Engelhardt Institute of Molecular Biology, Russian Academy of Sciences, 119991 Moscow, Russia; 3Federal Research and Clinical Center of Specialized Medical Care and Medical Technologies, Federal Medical and Biological Agency of Russia, 115682 Moscow, Russia

**Keywords:** glioblastoma, immunotherapy, CAR T, oncolytic viruses, poliovirus, viral vector, recombinant strain, cancer immunotherapy, nanoparticles, proteomics

## Abstract

Glioblastoma is the most aggressive form of malignant brain tumor. Standard treatment protocols and traditional immunotherapy are poorly effective as they do not significantly increase the long-term survival of glioblastoma patients. Oncolytic viruses (OVs) may be an effective alternative approach. Combining OVs with some modern treatment options may also provide significant benefits for glioblastoma patients. Here we review virotherapy for glioblastomas and describe several OVs and their combination with other therapies. The personalized use of OVs and their combination with other treatment options would become a significant area of research aiming to develop the most effective treatment regimens for glioblastomas.

## 1. Introduction

Glioblastoma remains one of the most aggressive and incurable malignancies [1]. Gliomas represent 30% of primary brain tumors and 80% of all malignant ones [2]. Gliomas are classified by the World Health Organization (WHO) into four grades, with glioblastoma categorized as grade IV [3]. These grades indicate a different degree of malignancy and help to select the proper treatment for each case [4]. In 2021, WHO updated its classification of central nervous system (CNS) tumors in which genetic features and molecular patterns alongside histopathological ones are used in designating different kinds of gliomas [1,5]. Glioblastoma is recognized as an IDH-wildtype diffuse glioma and the “IDH-mutant glioblastoma” term was eliminated [5]. This review uses the abbreviation “GB” to indicate glioblastoma instead of “GBM”, originally used for glioblastoma multiforme. GB is the most aggressive glioma, accounting for about 50% of all cases [6]. The median overall survival for GB ranges from 14.6 to 20.5 months, with less than 5% of patients surviving five years after diagnosis [7,8]. The standard treatment for GB involves a surgical resection followed by radiation therapy and chemotherapeutic temozolomide (TMZ) [9]. Along with the TMZ, four drugs and one medical device are approved by FDA for GB treatment: bevacizumab (BVZ), lomustine, intravenous carmustine, carmustine wafer plants, and tumor ultrasound fields [10]. Despite different treatment approaches, GB still has one of the lowest overall survival rates in all cancer types. No identified causes for developing GB have been established so far. Some risk factors such as a high dose of radiation [11], obesity [12], human cytomegalovirus (CMV) [13], and family cancer history [14] are among the ones to be studied. The molecular characteristics of GB include high intratumoral heterogenicity [15], tumor-induced immunosuppressed microenvironment [16], low infiltrating immunity [17], presence of stem-like cells called glioma stem cells (GSCs) [18,19], the existence of the blood–brain barrier (BBB) [20], and low tumor mutational burden in GB [21,22]. Advances in molecular pathogenesis are translated today into a more accurate characterization of the GB microenvironment, including genomic, epigenomic, transcriptomic, and proteomic characterization and its interactions with the immune system [23]. Therefore, novel therapeutic strategies are urgently needed to surpass the current obstacles in GB therapy. Oncolytic viruses (OVs) are among the most promising treatments for GB and brain tumors [24]. At the beginning of the nineteenth century, several studies have shown that viral infections can lead to the regression of neoplastic lesions [25]. In the early twentieth century, a woman with acute leukemia showed clinical remission after a viral infection [26]. Active exploration of oncolytic virotherapy started in the 1990s with recombinant DNA technologies for virus genome engineering. Herpes Simplex Virus (HSV) was the first genetic modification designed to selectively destroy GB cells in vitro and in vivo [27]. Several ongoing clinical trials evaluate the potential of different OVs in GB treatment [28,29]. OVs use various mechanisms to destroy cancer cells and amplify their therapeutic effects.

Several immunotherapy combinations with oncolytic viruses are developing to foster outcomes and minimize adverse effects [30]. For example, immune checkpoint inhibitors (ICIs) [31], Chimeric Antigen Receptor T therapy (CAR-T) cells [32], vaccine therapies [33], focused ultrasound therapy [34,35], and novel bioengineered nanoparticles [36] are all in the pipeline for GB treatment.

In this review, we discuss the status of oncolytic viruses in GB therapy and other combination therapies that are developing today. Also, we look through recent progress in preclinical and clinical trials and future perspectives for emerging developments. 

## 2. Oncolytic Viruses in GB

OVs are suitable for GB treatment due to their characteristics: alignment to the brain environment, no distant metastases, and fast-growing cells that attract virus replication [37]. The antitumor immune response initiates by turning the immunosuppressive microenvironment “cold tumors” into immune-responsive “hot tumors” [38]. Immunogenic cell death (ICD) is characterized by inducing an immune response to indirectly kill the cancer cells using different mechanisms such as apoptosis, necrosis, and autophagy [39]. This is represented by releasing tumor-associated antigens (TAAs), damage-associated molecular patterns (DAMPs), viral pathogen-associated molecular patterns (PAMPs), and several other cytokines [40]. Oncolytic viruses facilitate the function of antigen-presenting cells (APCs), which migrate to the lymph nodes to activate the cytotoxic CD8+ T lymphocytes (CTLs) and recruit them to the infection site leading to tumor cell killing [41] (Figure 1). 

The most frequent immune cells in the glioblastoma tumor microenvironment (TME) are macrophages that originate from peripheral-derived monocytes and are called tumor-associated macrophages (TAM) [42,43]. TAMs express mainly surface marker M2 and produce IL-10 and TGF-β with STAT3 expression to favor immunosuppressive condition “cold” tumors in glioblastoma [44]. Also, the exhaustion of T cells with upregulating inhibitory molecules and increasing Treg cells are among the characteristics of glioblastoma [45]. Furthermore, recent studies show that several chemokine expressions, such as CXCL2, CX3CL1, CCL5, and CCL2, are significantly higher in glioblastoma [46]. Hence, oncolytic viruses can benefit the therapy by reversing these conditions in favor of more immune infiltrating and better tumor killing [47].

Today, virotherapy is considered a promising type of immunotherapy for GB. OVs are classified into two groups: (1) replication-competent OVs that selectively replicate in cancer cells; and (2) replication-deficient viral vectors used as vehicles for other therapeutic genes. The first group can be divided into naturally occurring and genetically engineered viruses. The first are parvoviruses, reoviruses, and Newcastle disease viruses (NDV). Adenoviruses (Ad), HSV, vaccina viruses (VV), vesicular stomatitis viruses (VSV), polioviruses, and measles viruses (MV) are vulnerable to genetic manipulation that reduces their pathogenicity and increases their tumor selectivity. Specific OVs use selective receptors expressed on tumor cells to allow virus replication and further induction of the antitumor immune response. 

Over 20 oncolytic viruses have been tested in clinical trials for GB treatment. Among them are HSV-1 [48,49,50], Ad [51], Reovirus [52], MVs [53,54], NDVs [55], and poliovirus [56] (Table 1). In addition, new techniques in OVs delivery are improved to avoid the BBB limitation, as in the convection-enhanced delivery (CED) of the recombinant nonpathogenic polio-rhinovirus chimera (PVSRIPO) [28]. CED is a sophisticated new technique that uses a pressure gradient in a catheter to transfer therapeutic compounds in the interstitial spaces of the CNS [57]. The efficient and safe delivery of oncolytic viruses is crucial for successful virotherapy. The challenge of delivering viruses to the CNS and the elimination by the immune system led to choosing intratumoral delivery as the primary way [58]. Nevertheless, the oncolytic virus ideally should be delivered systematically to infect primary and metastatic tumor sites [59]. Fortunately, glioblastoma has a low frequency of metastatic spread incidence outside CNS [60].

Several factors should be accounted for when choosing a method of delivery. That includes whether the virus can cross the BBB and its immune clearance, the feasibility of intratumoral delivery, and the dose limit of the virus. The challenge of innovating new methods that can deliver therapeutic effects to distant tumor sites compared to injected ones would be critical to the success of virotherapy in the future. Thus, several studies are testing new modifications to use biological vectors and other bioengineering methods to enhance virotherapy efficiency. Thus, alternative approaches that include various biological vectors and bioengineering methods are being tested to improve the therapeutic delivery of OVs to tumor sites. For example, neural stem cells (NSCs) have been used to deliver the oncolytic Ad (NSC-CRAd-S-pk7) in phase I clinical trials for glioma patients. This method was safe and effective, with minimal dose-limiting toxicity [61]. In addition, lymphocytes can be thought to be used as vectors for oncolytic viruses after altering their characteristics with immortalization by herpes virus saimiri (HVS) [62]. Thus, the emergence of new treatments is a life necessity. The permanent development of OVs against GB improves their characteristics and provides effective agents to be tested in clinical trials (Table 1).

### 2.1. DNA Viruses

Herpes Simplex Virus Type I

The HSV-based oncolytic virus was the first designed virus strain tested for experimental therapy in a murine GB model [27]. HSV-1 is a double-stranded DNA virus and a member of the *Herpesviridae* family. HSV-1 attaches to cell surface protein CD111(nectin-1), which is highly expressed in GB [63]. The granulocyte-macrophage colony-stimulating factor (GM-CSF)-expressing talimogene laherparepvec (T-VEC) got the approval of the US food and drug administration (FDA) in 2015 to treat unresectable advanced melanoma [64,65,66]. Several modified HSV constructs are undergoing clinical trials in patients with GB, including G207, HSV-1716, M032, and MVR-C252 (Table 1). Both copies of the *RL1* gene encoding the virus protein (ICP34.5 inducing neurovirulence) were deleted in all HSV recombinants [67], leading to enhanced tumor selectivity [68]. The G207 recombinant variant of HSV with broken viral ribonucleotide reductase (RR) could not replicate in normal cells. Tumor cells compensate for the loss of RR by synthesizing the homologic gene [69]. The G47Δ strain represents a third-generation oncolytic HSV-1 (named DELYTACT), which demonstrated promising results in phase II clinical trials [70]. G47∆ was developed by adding deletion modification to the ICP47 gene to the enhance anti-tumor response via immune recognition using MHC class I [71]. G47Δ was initiated in a phase 2 single-arm trial for 19 adult patients with residual or recurrent glioblastoma. G47Δ was administered intratumorally in up to six doses. The defined endpoint was met with a one-year survival rate of 84.2%. Fever was the most common adverse event in 17 out of 19 patients. Tumor-infiltrating CD4^+^/CD8^+^ lymphocytes were observed in the patients’ biopsies [72]. Another phase I/II study of G47Δ in recurrent/progressive glioblastoma patients indicated a median overall survival of 7.3 months with 38.5% having a 1-year survival rate [73]. These studies led G47Δ to be granted conditional approval in Japan in 2021. Tumor-specific promotors (such as Nestin-1) are used in recombinant virus rQNestin34.5, controlling ICP34.5 expression for enhanced cytolytic activity in tumor cells [74]. M032 is a recombinant HSV expressing human interleukin 12 (IL-12) to increase interferon-gamma (IFN-γ) production and antitumor activity [75,76]. NG34 is a new oncolytic HSV (oHSV)-developed form of rQNestin34.5 with deletions in the ICP6 and ICP34.5 genes. NG34 revealed a similar effectiveness to its predecessor though with less toxicity in vivo [77]. rRp450 is another oHSV that contains a deletion in ICP6 and an insertion of CYP2B1 to enable activating the prodrug cyclophosphamide (CP) [78]. This virus improved survival in tumor-bearing mice with enhanced efficacy after adding the CP agent. OV-CDH1 is an engineered HSV expressing E-cadherin to increase the virus spread in the tumor by increasing the oncolytic effect and inhibiting the NK-mediated immunity in the infected cells [79,80]. Another oHSV expresses matrix metalloproteinase 9 (MMP) targeting tumor-specific EGFRvIII mutant antigen and enhances the tumor spread [81]. Also, this virus bears a specific recognition site for miR-124 to inhibit important virus protein ICP4 in normal glial cells and improve its specificity against tumor brain cells [82]. The oHSV that expresses Flt3L has revealed a complete GB clearance in preclinical studies [83]. Another oHSV expressing TRAIL, a protein that activates TNF–CD95L and induces apoptosis, exhibited cytotoxic activity in GB models in mice with prolonged survival rates [84]. oHSV armed with PD-1 antibody showed curable effects against GB mouse models [85].

Adenovirus

Adenoviruses are double-stranded DNA viruses with icosahedral non-enveloped structures [86]. Several versions of conditionally replicative adenoviruses (CRads) were produced and have given promising results in clinical practice against GB [87]. Additionally, a genetically engineered adenovirus is currently undergoing clinical trials in patients with GB alone or combined with other ICIs (Table 1). Another approach in using gene-mediated cytotoxic therapy was reported to be safe and beneficial in malignant glioma [88]. The phase II clinical trial was conducted using adenovirus glatimagene besadenovec (AdV-tk), which contains the HSV thymidine kinase gene that destroys cancer cells after acting on alacyclovir [89]. Deletion of the viral replication genes is one strategy to avoid off-targets in normal cells and can still replicate in tumor cells. H101 (Oncorine), similar to oncolytic adenovirus ONYX-015, was approved in China for the treatment of head and neck cancer [90,91]. A deletion in the *E1B-55K* gene in the recombinant adenovirus ONYX-015 oncolytic restricted viral replication to the p53-defective tumors [92]. Two genetic modifications were introduced in DNX-2401, a serotype 5 Ad (Ad5)-based OV [93]. A deletion in the *E1A* gene and inclusion of an RGD-4C motif in the HI loop of the fiber redirects virus replication to cells with defective pRB pathways that express αvβ3- and αvβ5-integrins, both of which are characteristics for glioma cells [94]. The modification enables adenoviruses to enter cells, even with low levels of their primary receptor on brain tumor cells, the coxsackie-adenovirus receptor [95]. The second generation of DNX-2401, DNX-2440 (or Delta-24-RGDOX), expresses the *OX40L* gene to improve T cell-mediated immunity by increasing the proliferation of CD8+ specific-tumor T cells [96,97]. Delta-24-ACT is an oncolytic adenovirus expressing 4-1BB ligand to further stimulate T cells in murine glioma models [98]. Delta-24-RGD is another oncolytic adenovirus armed with glucocorticoid-induced TNFR family-related gene ligand (GITRL) that enhanced the survival and inhibited further rechallenge with glioma cells in mice [99]. Also, using another oncolytic adenovirus (Delta-24-RGDOX) expressing co-stimulator OX40 ligand (OX40L) resulted in CD8+ T-cells proliferation and cancer-specific immunity in vivo [100].

Vaccinia Virus (VV)

Vaccinia is a double-stranded DNA virus belonging to the Poxviridae family. VV helped in the eradication of smallpox. VV can infect any type of cell as it penetrates through membrane fusion with a non-integrative replication cycle, making it an attractive platform for oncolytic virus engineering against GB [101]. The only recombinant VV that showed clinical benefits in patients with brain malignancies is TG6002 [102]. Two gene deletions of the thymidine kinase (TK) gene and the RR gene were introduced in the TG6002 genome. Also, the *FCU1* gene was inserted to convert the chemotherapeutic prodrug 5-flucytosine (5-FC) into 5-fluorouracil (5-FU) [103].

Myxoma virus

Myxoma virus (MYXV) is a double-stranded DNA member of the poxvirus family [104,105]. MYXV replicates in the cells with a disabled interferon system, such as GB, where it can induce an oncolytic effect [105]. The M011L-deficient MYXV virus, which has a deleted version of the viral antiapoptotic protein M011L, increased apoptosis in tumor glioma cells [106].

Parvoviruses

Parvoviruses are single-stranded icosahedral DNA viruses that belong to the Parvoviridae family. About 134 distinct parvoviruses serotypes can infect various animal species [107]. H-1 parvovirus is a minor oncolytic virus that showed antitumor activity against GB [108]. In addition, H-1PV induces apoptosis in glioma cells and overcomes their resistance against several chemotherapeutic agents [109]. Preclinical findings of the H-1PV revealed tumor regression in human U87-MG glioma models in rats [110]. This led to the initiation of the ParvOryx01 trial in recurrent GB patients (NCT01301430). ParvOryx01 indicated the role of tumor-infiltrating lymphocytes (TILs) in stimulating immune effects in the resected tumor tissues of GB patients [111]. Radiation increases H-1PV viral oncolysis in high-grade human gliomas and can potentially be considered in animal glioma models [112]. A combination of bevacizumab and H-1PV extended the mean survival to 15.4 months in five recurrent GB patients and caused remission in three patients [113]. These effects are related to the synergistic effect of H-1PV and bevacizumab in modulating GB TME and inhibiting the vascular endothelial growth factor (VEGF) [114]. The first clinical evidence of using H-1PV with an immune checkpoint inhibitor (nivolumab) and bevacizumab was reported in the multimodal clinical trial of three recurrent GB patients. Tumor regression and clinical improvement were documented in all subjects, with 78% of cases showing complete or partial remission [102]. Altogether, these data indicate the potential of parvoviruses for combination immunotherapies against GB.

### 2.2. RNA Viruses

Measles Virus

Measles virus (MV) is a negative-sense, single-stranded RNA virus and a member of the Paramyxoviridae family [103]. The MV enters cells by interacting with the viral hemagglutinin (H) protein and the CD46 cell receptor overexpressed on tumor cells [103]. Recombinant MVs showed significant antitumor activity in glioma xenografts and entered clinical trials [104,105]. Such recombinants express human carcinoembryonic antigen (CEA) or the human sodium iodide symporter (NIS) to track viral expression in cells [106]. NIS permits virus monitoring using different isotopes and could be used to increase virus cytopathic effects [107,108].

Vesicular Stomatitis Virus (VSV)

The VSV is a negative-sense, single-stranded RNA virus and a member of the Rhabdoviridae family. The VSV uses the connection between its spike glycoprotein (G) and the low-density lipoprotein receptor (LDL-R), which is a ubiquitous cell receptor [109]. The VSV replicates in tumor cells using the aberrations in their interferon system and is used as an oncolytic agent against several tumors [110,111]. rVSV(GP) and VSV-EBOV are engineered VSV with the envelope glycoprotein (GP) replaced with GP from the non-neurotropic lymphocytic choriomeningitis virus and Ebola virus, respectively [112,113].

Reoviruses

Reoviruses are double-stranded RNA non-enveloped viruses that can replicate in the glioma cells with the activated Ras-signaling pathway [114].

Newcastle Disease Virus (NDV)

NDV is a negative-sense, single-stranded RNA-enveloped virus and a member of the Paramyxoviridae [115]. The NDV is mainly an avian virus that preferentially replicates in tumor cells and triggers the type I interferon response by expressing interferon-stimulated genes (ISGs) in humans [116,117]. Studies show that NDV may be effective against GB [118].

Seneca Valley Virus Isolate 001 (SVV-001)

The SVV-001 is a positive-sense single-stranded RNA member of the Picornaviridae family [119]. The SVV-001 has exhibited oncolytic activity against solid tumors with selective tropism to the cells expressing the endothelial receptor TEM8/ANTXR1 [120]. Enriched in several cancer types, TEM8/ANTXR1 is an integrin-like, transmembrane glycoprotein adhesion molecule that meditates cell movement and its interactions with the extracellular matrix (ECM) [121]. TEM8/ANTXR1 represents a first biomarker oncolytic viral therapy using SVV [122]. SVV-001 can cross the BBB intravenously and provide antitumor activity [123].

Polioviruses

Polioviruses are positive-sense single-strand RNA viruses and members of the Picornaviridae family [124]. Polioviruses infect cells by using the CD155/PVR receptor frequently over-expressed on malignant cells [125].

PVSRIPO is based on an attenuated poliovirus type 1 (Sabin) vaccine strain in which its internal ribosome entry site (IRES) was replaced by IRES from the human rhinovirus type 2 to restrict the potential neurovirulence [126,127]. The phase I trial (NCT01491893) investigating the intratumoral CED of PVSSRIPO in patients with recurrent GB proved the safety and the absence of neurovirulence. Consequently, the PVSRIPO was granted a breakthrough therapy status by FDA in 2016 [28]. Furthermore, the data reveal that the survival rate in this trial was higher, at 24 and 36 months, reaching 21% higher than the rate among the historical controls. A phase II trial (NCT02986178) of PVSRIPO alone or in combination with lomustine is going on in GB patients, and its results are much awaited [128].

RVP3 is a new recombinant poliovirus type 3 vaccine strain with IRES replaced by the human rhinovirus type 30 that replicates selectively in tumor cells without infecting normal cell lines [129]. RVP3 revealed oncolytic efficacy on different glioma models and primary glioma cells from different patients [129].

Sindbis virus

The Sindbis virus is an avian positive-sense single-stranded RNA virus and a member of the Togaviridae family [130]. Sindbis infects cancer cells by attaching to the laminin receptor (LAMR) [131] and induces apoptosis by activating the tyrosine phosphorylation of protein kinase C delta in glioma cells [132].

SFV4miRT is the Semliki Forest Virus with inserted target sequences for miR124, miR125, and miR134 expressed more in normal CNS cells than in glioma cells [133]. Thus, this virus has reduced neurotropism with oncolytic efficacy and a safer profile [134]. Recently, studies have shown that the Zika virus (ZIKV) can infect the GB stem cells (GSCs) and have oncolytic activity on them [135,136]. That suggests that engineering ZIKV to target GB more selectively without normal neuronal cells can give better therapeutic results [137,138]. A 10-nucleotide deletion in the 3′ untranslated region of the genome (3-UTR) resulted in ZIKV-LAV that has better oncolytic efficacy against GB with less neurovirulence [139,140].

In general, most clinical trials with oncolytic viruses for GB proved the safety and efficacy of OVs on glioma cells, but few of them proceed to phase III. Sitimagene ceradenovec, an adenoviral vector encoding the HSV’s thymidine kinase gene followed by intravenous ganciclovir, was evaluated in the phase III clinical trial “ASPECT”. However, no significant effect on overall survival was noticed [141]. Toca511, which consists of the retroviral vector with cytosine deaminase gene (CD), has entered a phase III trial. The CD gene in Toca511 converts the 5-flucytosine to the cytotoxic drug 5-fluracil to kill cancer cells [142]. Also, this trial was ended for unclear reasons. It is worth noting that despite the safety and efficacy of OVs in preclinical, the clinical effectiveness has not reached the promised level.

## 3. Oncolytic Viruses with Immunotherapy in GB

The most frequent immune cells in the GB tumor microenvironment (TME) are macrophages that originate from peripheral-derived monocytes and are called tumor-associated macrophages (TAM) [143,144]. TAMs mainly express surface marker M2 and produce IL-10 and TGF-β with STAT3 expression to favor immunosuppressive condition “cold” tumors in GB [145]. Also, the exhaustion of T cells with upregulating inhibitory molecules and increasing Treg cells are among the characteristics of GB [146]. Recent studies show that the expression of immunomodulatory chemokines CXCL2, CX3CL1, CCL5, and CCL2 is high in GB [136]. Hence, oncolytic viruses can benefit the therapy by reversing these conditions in favor of more immune infiltrating and better tumor killing [147]. Immune checkpoint inhibitors (ICIs) and CAR-T therapies represent the main mechanisms against GB. Immune checkpoints, in general, exist to inhibit further immune responses and undesired immune reactions in a so-called “self-guarding” way to maintain a balanced immune response. Unfortunately, tumors such as GB can use this mechanism to evade immune surveillance by increasing immune checkpoints such as PD-1, PD-L1, IDO, and CTLA-4. The primary role of immunotherapy in GB is to change its “cold” tumor environment into a “hot” one that can react to different therapeutic applications. One study showed that combining the CTLA-4 blocker and IL-12-induced T cell mediated glioma rejection in a GB murine model [148]. However, no single ICI demonstrated an apparent clinical effect in the phase III trials, and the FDA has not issued any approval [32].

Given that most ICIs work in tumors with a high mutational burden [149], GB with a low mutational burden represents a significant challenge to immune checkpoint blockade therapy. That explains the need to identify next-generation immune checkpoints and better characterize the GB biomarkers, including their mutational burden. Also, new strategies for using oncolytic viruses with ICIs should be developed. The newly discovered cluster of differentiation 47 (CD47) sends inhibitory signals to the innate immunity and protects the tumor from macrophage attack [150]. Several studies examined the CD47/SIRPα axis blockade for GB tumors by targeting CD47 [149]. The Anti-CD47 antibody stimulated phagocytosis and induced microglia remodeling in GB tumors in vivo [151]. In addition, the CD47 blockade induced phagocytosis via macrophages, enhancing the overall survival in human GB models in mice [152].

Additionally, anti-CD47 immunotherapy could remodel the GB microenvironment and can be combined with other treatments such as OVs, irradiation, and chemotherapy [153]. CD73 is considered a new target for the immune checkpoint as it degrades adenosine monophosphate (AMP) to adenosine (ADO) that mediates immunosuppression [154]. Preclinical studies of the CD73 blockade in GB models decrease the GB growth and modulate the microenvironment [155]. The data from several GB immune profiles identify CD73 as a specific target for human GB therapy [156]. Although all these findings are promising, it remains to be seen whether the preclinical effects of CD47 and CD73 will be the same in clinical trials on GB patients [157]. 4-1BB (CD137) is a costimulatory receptor that triggers T and NK immune cell proliferation and activation [158,159]. Using anti-CD137 agonists mediates antitumor activity and can be used with other monoclonal antibodies to induce an antitumor response [158,160]. The combination of new checkpoints targeted with oncolytic viruses seems to have a promising prognosis, as OVs can induce remodeling of the GB immunosuppressive environment.

Using OVs with immune checkpoint inhibitors (ICIs) has given promising results in several studies [161]. MV infection in GB models increased the expression of the PD-L1 molecule [162]. Thus, using both MV and anti-PD-1 antibodies resulted in a better survival rate in mice glioma models than in each therapy alone [163]. Another study indicated an enhanced survival rate in GL261 tumor-bearing mice after intravenous infusion of reovirus expressing GM-CSF followed by anti-PD-1 antibodies [164]. Also, oncolytic adenovirus DNX-2401 with anti-PD-1 significantly altered the tumor microenvironment and enhanced the survival of both GL261 and CT2A murine glioma models [165]. Combination therapy of miR-124 oHSV with anti-PD-1 demonstrated an antitumor immune response in the GB model [166]. The synergistic effect of oHSV expressing IL-12 with the two ICIs, the anti-PD-1 and anti-CTLA-4 antibodies, was assessed in a triple-combination therapy [167]. This approach proved its effectiveness in reversing the immunosuppressive TME of GB and eradicating GSCs [168]. Another study using VSV encoding three TAAs, HIF-2α, Sox-10, and c-Myc, with two ICIs, the anti-PD-1 and anti-CTLA-4 antibodies, turned out to be more effective than using VSV alone or with either ICI alone [169]. A phase II clinical trial of oncolytic virus DNX-2401 in combination with the anti-PD-1 antibodies (pembrolizumab) indicated a 100% survival rate at nine months in the first treated patients [170]. The parallel use of the double-deleted Vaccinia virus (vvDD) or MYXV expressing the IL15Rα-IL15 fusion protein with other combinations such as rapamycin, celecoxib, and specific glioma neoantigen (GARC-1), resulted in a higher efficacy in GL261 murine glioma models [171]. OVs can synergize with CAR T-cell therapy to alter the TME of glioma cells and enhance T-cell infiltration and effective functions. Several studies revealed the success of such a strategy in different solid tumors [172,173]. However, a recent study of VSV encoding interferon beta (IFNβ) showed a reverse effect on CAR-T cells targeting EGFRvIII in B16 murine tumors [174]. Such a result leads to taking into consideration further optimization to understand all the immunological aspects of this combination [175]. Of note, such a combination is still to be tested in GB as it possibly would enhance the efficacy of the treatment. Several combinations of OVs with the chemotherapeutic drug TMZ revealed enhanced survival rates [176,177,178]. The Oncolytic virus Delta-24-RGD increased CD8^+^ T cell infiltration in the presence of TMZ, improving overall survival in the murine GL261 model [179]. Toca 511 and TG6002 OVs can convert 5-FC prodrug into cytotoxic 5-FU in clinical trials [180]. The Oncolytic virus vvDD expressing enhanced green fluorescence protein (EGFP) in combination with rapamycin or CP prolonged the survival rate in glioma models of immunocompetent mice [181]. Recently, the clinical potential of CAR-T in GB therapy was examined using targets of ephrin-A2 (Her2), interleukin 13 receptor alpha-2 (IL-13Rα2), and the epidermal growth factor receptor variant III (EGFRvIII) [182,183,184]. The first promising results were obtained by specific CAR-T cells against IL-13Rα2 in GB treatment [185]. Tumor regression and increased levels of immune response (cytokines and infiltrating immune cells) were observed in the tumor region [186]. Another study revealed that the infusion of HER-2-CAR autologous virus-specific T cells (HER2-CAR-VSTs) in GB patients resulted in safe and clinically beneficial progress [187]. No survival benefit was observed when using EGFRvIII-CAR-T cells in recurrent GB patients expressing EGFRvIII, even with the secure treatment profile [188]. GB heterogenicity and antigen escape are the primary mechanisms for avoiding CAR-T therapy, as lower expressions of previous antigens were noticed [189]. Bispecific T cell engagers (BiTEs) were proposed to reduce these obstacles by using a bi-targeted platform, which connects T cells to tumor cell-specific antigens [179]. BiTEs prevented antigen escaping and indicated antitumor activity in GB treatment [190]. In summary, thoroughly understanding each treatment mechanism is essential in combining different therapies to choose the most convenient ones. Selecting the best OVs in parallel with the best delivery method is crucial to address each cell type and consider a personalized approach here.

### Novel Pre-Clinical Oncolytic Viruses in GB

Few new developments have led to approved drug therapies for glioblastoma [190]. The permanent development of new OVs against glioblastoma aims to improve their characteristics and provide the most promising candidates to be tested in clinical trials. NG34 is a new oHSV derivate of rQNestin34.5, in which a double deletion in the ICP6 and ICP34.5 genes is introduced. NG34 revealed a similar effectiveness to its predecessor with less toxicity in vivo [77]. rRp450 is another oHSV that contains a deletion in ICP6 and an insertion of CYP2B1 to enable activating the prodrug cyclophosphamide (CP) [78]. This virus improved survival in tumor-bearing mice with enhanced efficacy after adding the CP agent. OV-CDH1 is an engineered HSV expressing E-cadherin to increase tumor spread by protecting infected cells from NK-mediated lysis [79,80]. Another oHSV-expressing matrix metalloproteinase (MMP) 9 protein that is tumor-specific, EGFRvIII, is a mutant antigen and enhances tumor spread [81]. Also, this virus bears a specific recognition site for miR-124 to inhibit important virus protein ICP4 in normal glial cells and improve its specificity against brain tumor cells [82]. oHSV that expresses Flt3L revealed complete glioblastoma clearance in preclinical studies [83].

Another oHSV expressing TRAIL, a protein activating TNF–CD95L and causing apoptosis, exhibited cytotoxic activity in glioblastoma models in mice with a prolonged survival rate [84]. oHSV armed with the PD-1 antibody showed curable effects against glioblastoma mouse models [85]. Delta-24-ACT is an oncolytic adenovirus expressing the 4-1BB ligand to further stimulate T cells in murine glioma models [98]. Delta-24-RGD is another oncolytic adenovirus armed with glucocorticoid-induced TNFR family-related gene ligand (GITRL) that enhanced the survival and inhibited further rechallenge with glioma cells in mice [99]. Also, using another oncolytic adenovirus (Delta-24-RGDOX) expressing the co-stimulator OX40 ligand (OX40L) resulted in CD8+ T cell proliferation and cancer-specific immunity in vivo [100]. The M011L-deficient MYXV virus, which has a deleted version of the viral antiapoptotic protein M011L, increased apoptosis in tumor glioma cells [106]. rVSV(GP) and VSV-EBOV are engineered VSV with the envelope glycoprotein (GP) to be replaced with GP from the non-neurotropic lymphocytic choriomeningitis virus and Ebola virus, respectively [125,126]. SFV4miRT is a Semliki Forest virus with inserted target sequences for miR124, miR125, and miR134 that are more expressed in normal CNS cells than in glioma cells [146]. Thus, this virus has reduced neurotropism with oncolytic efficacy and a safer profile [136]. Recent studies have shown that the Zika virus (ZIKV) can infect and destroy glioblastoma stem cells (GSCs) [147,148]. This suggests that engineering ZIKV to target glioblastoma more selectively without normal neuronal cells can give better therapeutic results [149,150]. A 10-nucleotide deletion in the 3′ untranslated region (3-UTR) resulted in engineering ZIKV-LAV, which is less neurotoxic and has better oncolytic efficacy against glioblastoma [149,151].

In general, most clinical trials with oncolytic viruses for glioblastoma confirmed the safety and efficacy of OVs against glioma cells, but few proceed to phase III. Sitimagene ceradenovec, an adenoviral vector encoding the herpes simplex thymidine kinase gene followed by intravenous ganciclovir, was evaluated in the phase III clinical trial “ASPECT”. However, no significant effect on overall survival was noticed [152]. Toca511, which consists of a retroviral vector with cytosine deaminase gene (CD,) has entered a phase III trial. The CD gene in Toca511 converts the 5-flucytosine to the cytotoxic drug 5-fluracile to kill cancer cells [153]. However, this trial was ended for unclear reasons. It is worth noting that despite the safety and efficacy of OVs in preclinical studies, the clinical effectiveness has not reached the promised level.

## 4. Other Therapeutic Approaches in GB

### 4.1. Targeting GB-Specific Antigens

Cancer vaccines emerged as a new promising therapy [191]. The most studied antigen in GB is EGFRvIII. A mutant EGFR is expressed in 25–30 % of GB cases [192]. A peptide vaccine named rindopepimut (or CDX-110) indicated improved median survival of 24 months after targeting EGFRvIII in a phase II trial for GB patients [193]. A phase III was launched based on previous findings, but no survival benefit was detected compared to the control group [194]. The reason behind such results could be antigen loss and heterogenous expression [195]. Recently, another study showed the effectiveness of rindopepimut in EGFRvIII-positive GB patients in a double-blind, randomized phase II trial [196]. ICT107 is a dendritic-cell (DC) vaccine against GB antigens in HLA-A2-positive GB patients that led to beneficial outcomes [195]. Another DC vaccine, DCVaxL, which uses tumor lysate to induce patients’ DCs, gave promising results regarding the median overall survival (OS) rate in GB patients [8]. However, the results of a large phase III clinical trial of an autologous dendritic cell vaccine had a median OS of 23.1 months and was considered superior to the 15–17 median OS in previous trials [33]. Again, the absence of specific GB antigens and the high rate of heterogenicity hinders the development of more effective vaccines against GB.

Nonetheless, advances in bioinformatics and sequencing methods led to identifying neoantigens, tumor-specific antigens derived from somatic tumor mutations. Neoantigens target personalized antigens in cancer patients to initiate robust T-cell response and anti-tumor activity [197]. A study on MGMT-unmethylated GB patients showed an increase in tumor-infiltrating T cells after vaccinating with neoantigens [198]. Unfortunately, the immune response did not last, and tumor recurrence was observed, revealing clinical obstacles in the tumor microenvironment [198]. In summary, current vaccines are insufficient to treat GB, given tumor heterogenicity and immunosuppressive factors. Thus, combining vaccines with other therapies such as oncolytic viruses may be an excellent solution to alter the defects in the immune response against GB.

### 4.2. Nanoparticles for GB Treatment

Nanomedicines are therapeutic formulated materials used in different immunotherapy applications [199]. Nanoparticles (NPs) can enhance GB therapeutic efficacy using their distinctive properties in increasing bioavailability and accumulation in the TME [200,201] (Figure 2). NPs can be used as carriers for immunotherapeutic agents such as small interfering RNAs (siRNAs), neoantigens, and immune adjuvants [202,203]. Also, these compounds’ release can be controlled through NPs using their surface-to-volume ratio, surface modification, and drug-loading capacity [204]. Lipid-based nanoparticles are compatible with delivering hydrophobic and hydrophilic targeting agents due to their bilayers of phospholipids structure [205]. In several studies, liposome NPs indicated a therapeutic effect against GB [201,202]. Applying mannosylated lipid NPs resulted in tumor regression and macrophage phenotype shift from pro-tumorigenic phenotype (M2 phase) into anti-tumorigenic one (M1 phase) in glioma cells [206]. Another study showed that a liposome loaded with three different herbal compounds revealed antitumor activity and suppressing GSCs with repolarizing of TAMs [207]. Combining Paclitaxel (PTX) chemotherapy and immunotherapy using CpG material as an immune modulator was encapsulated in chitosan-coated lipid nanocapsules against GB models. Results exhibited an improved survival rate in mice compared to control groups and the only-treated-with-PTX group [208]. Lipid nanocapsules (LNCs) with a size of 100 nm efficiently target myeloid-derived suppressor cells (MDSCs) isolated from GB patients and induced their inhibition [209]. To notice the clinical significance of NPs is still to be discovered, but their therapeutic potential is promising.

### 4.3. Multi-Omics in GB

Several genetic and epigenetic abnormalities are involved in the molecular pathology of GB [210]. DNA methylation patterns differ from normal cells, such as the hypermethylation of MGMT promoter in about 40% of glioma cells [211]. Several studies showed the importance of miRNAs in controlling other mRNAs in GB tissues [212,213]. Inhibition of MiR-221/222 reduces tumor growth and malignant cell invasion in a xenograft glioma model [214]. A recent study showed that chromatin remodeling and histone modification could accelerate GB growth [215]. Thus, using novel epigenetic therapy in GB treatment is promising. Histone Deacetylase Inhibitors (HDACi) entered clinical trials to treat gliomas and recurrent GB [216]. The inhibition of the enhancer of Zeste homolog 2 (EZH2), which is H3K27 methyltransferase, resulted in the regression of GB primary cell cultures [217]. Developed proteomics using advances in mass spectrometry determines GB’s immunosuppressive patterns and protein heterogenicity [218]. Finding new biomarkers characteristic of GBs could stimulate the designing of new oncolytic viruses that target such unique, cancer-specific features [219]. Examining cell surface proteins or surfaceome expressions is appealing to find key-associated genes and proteins for GB treatment. A recent study identified 87 overexpressed surface proteins in GB models with five mutated proteins, such as RELL1, CYBA, EGFR, and MHC I [217]. In parallel with proteomics, RNA sequencing (RNA-seq) and single-cell RNA-seq enable us to determine associations between proteomic profiles and poor survival rates in different GB patients’ bulks [220]. Surfaceome analysis identified six molecular signatures upregulated in GB models: HLA-DRA, CD44, SLC1A5, EGFR, ITGB2, and PTPRJ [221]. Integrated metabolomic and proteomic data identify distinct histone H2B acetylation patterns and allow for a better understanding of the interrelated biological changes in GB models [222]. One study revealed five spatially distinct transcriptional programs that clarify the tumor-host interdependence in GB [223]. However, a fully comprehensive surfaceome and proteome landscape has not been fully elucidated. Thus, development in this direction will give potential candidates for new cell-surface antigens and other proteins as promising targets for new therapeutic purposes.

### 4.4. 3D Organoid Models in GB

The development of new therapies for GB faces several hurdles in developing preclinical models that resemble the tumor microenvironment (TME) [224]. Current models do not recapitulate the complex relationships between tumor cells and oncolytic viruses [225]. Thus, engineering three-dimensional (3D) organoid models increases preclinical GB models’ scalability and accuracy in testing new treatments [226]. Complex GB organoids (GBO) cultures were first gathered from multiple cell types derived from primary patient tumors to reflect their heterogenicity [227]. Neoplastic cerebral organoids (neoCOR) were developed via genetically manipulating cerebral organoids to induce GB growth [228]. Later, cerebral organoid GB co-cultures (GLICO) were generated from patient GSCs co-culturing with brain organoids to reflect the tumor–brain interactions [229]. Recently, bioprinted GB organoids were created as a GB-on-a-chip model that contains patient-derived multiple cells with various tumor gradients [230]. To achieve that, a decellularized porcine brain “bio-ink” composed of ECM proteins combined with GB cells from resected patients’ tumors. Also, such models were provided with human umbilical vein endothelial cells (HUVEC) to supply the organoids with important tumor features. Recently, other enhancements came to form a “tetra-culture” that incorporates macrophages and resembles immune interactions [231]. Culturing pluripotent stem cell-derived brain organoids with human GB stem cells (GSCs) in 3D maintains a significant cell fate heterogenicity and establishes a viable GB model [232]. A recent study indicated the GB organoid model reflects the entire tumor in practice. Both arms of the study using the patient-derived cells and organoids showed a similar molecular profile and resistance to a combination of TMZ and radiotherapy [233]. In general, organoids can be an alternative for the current 2D glioma models or the two in vivo models: the patient-derived xenografts (PDXs) and genetically engineered mouse models (GEMMs) [234]. Moreover, patient-derived GB organoids (PD-GBO) demonstrate representative results for molecular profiling to assign effective personalized treatment in GB patient models [235]. These organoids can help choose the appropriate oncolytic virus for each GB patient, depending on identifying biomarkers for specific OVs. Several models, such as organotypic tissues and slices to mimic the tumor microenvironment, are used in virotherapy. Individual patient “Virograms” is suggested for screening various OVs on the patient-derived organoid and choosing the ideal oncolytic agent [236]. One study suggested the 3D-direct evolution of several serotypes from adenovirus to be tested on patient organoids to get the most potent and effective viral isolate [237]. Nevertheless, establishing such models is laborious and needs cellular architecture and relations optimization. Problems in choosing the suitable model, reproducibility, cellular fate, and screening protocols have not been established yet. In the future, co-culturing GB cells after OV infection with immune cells may provide insights into the immune response in virotherapy.

## 5. Future Perspectives and Conclusions

As was described in this review, GB remains one of the most dangerous and malignant diseases. Oncolytic viruses are of significant promise in treating and activating the immune system against GB. Several considerations should be considered to get the most out of OVs. These are the OV’s ability to replicate and infect tumor cells, the status of the tumor microenvironment, the extent of immune cell infiltration, and the ability to induce an antitumor response. This review discussed the current aspects of GB treatment options using several approaches and specified oncolytic viruses currently used or tested in clinical trials. We also described novel OVs and several combination therapies (particularly immunotherapy) having potential therapeutic effects. Finally, we listed new techniques that may help oncolytic virotherapy to increase their impact and improve overall patient survival. Such approaches include applying nanoparticles in newly diagnosed patients, screening OVs in 3D-organoids resembling the patients’ tumors, and using multiomics analysis to understand the GB microenvironment better. Still, the future task in virotherapy would be to identify new therapeutic biomarkers and better adapt OVs to target these antigens in an individual–personalized manner. OVs can be the platform for designing new immunotherapeutic approaches, especially when using CAR-T cells and bi-specific T-cell engagers. Initial data from clinical trials for many ICIs and vaccine therapies were disappointing due to tumor heterogenicity. Thus, using and optimizing oncolytic viruses with such therapies seems to have considerable potential in GB treatment. Recent advances in personalized medicine revealed many neoantigens and shifts toward personalized virotherapy. This approach suggests having several choices of OVs for each patient to lower the tumor burden and improve the overall effect. Many OVs are in clinical trials today, and several considerations, including viral delivery and optimal dose, are under testing. Ex vivo personalized models may be developed to choose the most appropriate oncolytic virus for each patient. In future clinical trials, several OVs will be tested to correlate with tumor heterogenicity and immune status complexity. New monitoring and evaluation standards should be applied in virotherapy as novel approaches are tested. Future research should reveal the mechanism of OVs in different GB models using novel genetic engineering and viral delivery techniques. In brief, virotherapy as a standalone treatment may be effective, but combining strategies of immunotherapy and oncolytic viruses with the use of personalized approaches will pave the way toward a more effective treatment regimens for GB patients.

## Figures and Tables

**Figure 1 viruses-15-00547-f001:**
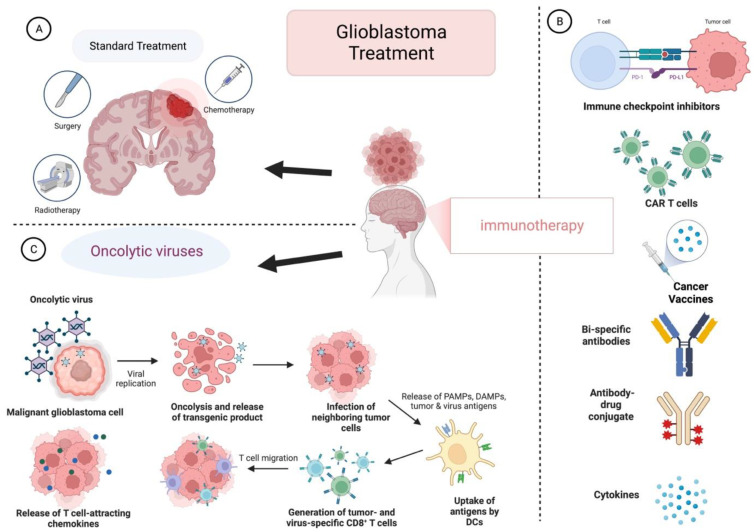
GB treatment types. (**A**) standard treatment regimens include surgical resection, chemotherapy, and radiotherapy. (**B**) Immunotherapeutic approaches include various types to activate the antitumor immunological response. (**C**) Virotherapy is promising because the oncolytic virus kills tumor cells using direct oncolysis and immunogenic cell death. The figure is created with the aid of BioRender.com.

**Figure 2 viruses-15-00547-f002:**
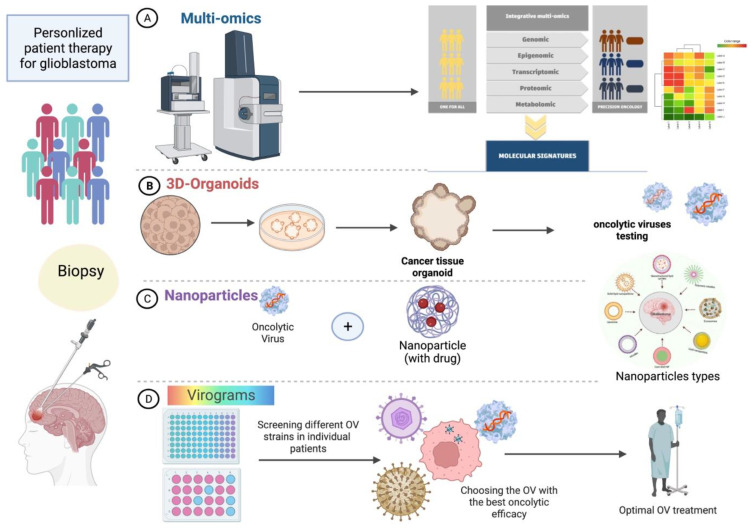
Personalized patient therapy for GB. In perspective trials for GB patients, personalized approaches should be used. Such approaches depend on the individual molecular characteristics of each patient. Analyzing patient biopsies in different platforms allows one to choose the optimal OV by synergizing its effect using new techniques. (**A**) Using multiomics to identify distinct molecular patterns and tumor microenvironment factors to use the optimal OV. (**B**) Using patient tumor cells to make 3D organoids and then testing several OVs on them to choose the appropriate OV for each patient. (**C**) Using nanoparticles to synergize the therapeutic effect of OVs in the patients. (**D**) Using new maps of “Virograms” to screen different OVs in individual patients and use the optimal variant. This leads to the stratification of the patients depending on their molecular patterns. The figure is created with the aid of BioRender.com.

**Table 1 viruses-15-00547-t001:** Complete oncolytic viruses in clinical trials for GB patients.

Oncolytic Virus	Status	Primary Outcome	NCT Number
DNA viruses			
Herpesvirus			
Genetically engineered HSV-1 MVR-C5252 (C5252)	Not yet recruiting phase I	Safety and tolerability DLTs and MTD	NCT05095441
Genetically engineered HSV-1 M032	Recruiting phase I	MTD	NCT02062827
A single dose of G207 infused through catheters into tumors	Recruiting phase I	Safety and tolerability	NCT03911388
Oncolytic viral vector rQNestin34.5v.2	Recruiting phase I	MTD	NCT03152318
Adenovirus			
Genetically engineered Adenovirus DNX-2440	Unknown phase I	Safety, OS, and ORR ^1^	NCT03714334
Adenoviral Nsc-crad-s-pk7	Phase I		NCT03072134
Adenovirus DNX-2401	Recruiting phase I	MTD and Incidence of AEs	NCT03896568
Parvovirus			
H-1 Parvovirus (H-1PV)	Completed phase I/II	Safety and tolerability	NCT01301430
RNA viruses			
Poliovirus			
Recombinant nonpathogenic polio-rhinovirus chimera (PVSRIPO) administered via CED into a tumor	Active, not recruiting phase I	MTD, DLTs, and RP2D	NCT01491893
PVSRIPO	Active, not recruiting phase I	Toxicity within 14 days after PVSRIPO treatment	NCT03043391
PVSRIPO administered via CED into a tumor	Active, not recruiting phase II	ORR rate and DORR at 24 and 36 months	NCT02986178
Reovirus			
Live, replication-competent wild-type reovirus REOLYSIN	Completed phase I	MTD, DLTs, and 6- month response rate	NCT00528684
Combinations of OVs			
Combination of modified vaccinia virus TG6002 and 5-FC	unknown phase I/II	DLTs and tumor progression at 6 months	NCT03294486

^1^ Most data were obtained from findings from www.clinicaltrials.gov (accessed on 3 November 2022) using the search terms “GB” and “oncolytic”; *OS* overall survival, *ORR* objective response rate, *IFN-γ* interferon Gamma, *SOC* Standard of Care, *DLT* dose-limiting toxicities, *AE* adverse event, *MTD* maximum tolerated dose, *HSV* herpes simplex virus, *CED* convection-enhanced delivery, *RP2D* recommended phase 2 dose, *ORR* objective radiographic response, *DORR* duration of objective radiographic response.

## Data Availability

Data are available upon request by contacting the corresponding author.

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
