# Peer review of "Recent Developments in Glioblastoma Therapy: Oncolytic Viruses and Emerging Future Strategies"

_viruses, 2023, doi:10.3390/v15020547_

Round 1

Reviewer 1 Report

This review introduces and discusses advances in the oncolytic virotherapy of glioblastoma, a deadly cancer with poor prognosis and limited treatment options.

The review starts by introducing the disease and the challenges of treating it. It then moves on to discussing OVs that have been developed and tested for the treatment, including DNA and RNA viruses. Finally, other approaches for the treatment of GB are discussed, including immunotherapy and the use of nanoparticles in the treatment of GB.

 Overall, this work is a valuable review of recent advances in the field of GB treatment. It is well written and is organized logically. My first comment is that the title indicates that OVs are to be discussed, but then, a large part of the review is dedicated to other therapeutic strategies. I suggest that the title be modified to indicate that the paper not only focuses on OVs, but rather discuss the broad strategies used to treat GB.

Here are some suggestions for this review:

1-    In line 73, you indicated that viruses induce antigen presenting cells (APCs) to the infection site to activate cytotoxic CD8+ T lymphocytes (CTLs) …”. This is not completely accurate. APCs pick up the tumor antigens, migrate to the LNs to activate the cytotoxic T-cells, which migrate back to the tumor site to induce tumor cell killing.

2-    Line 122: You can elaborate more on this method (convection-enhanced delivery (CED)), otherwise the reader has to go back to the reference to understand the concept.

3-    Line 125-126: Can you develop a rational classification for the delivery method of the virus for GB. e.g., systemic vs local, or with carrier vs naked virus.

4-      Sections 2.1-2.2: Please organize all viruses discussed in this review in a table, indicating the virus type and classification (DNA/RNA), whether it has been used in clinical trials (what phase). You can also indicate the references to each virus in this table.

Reviewer 2 Report

In this review Hamad and co authors have revised  the literature regarding the use of  Oncolytic viruses (OVs) for the treatment of Glioblastoma, the most aggressive form of malignant brain tumour.

The review is potentially relevant and timely, however I have several criticisms.

Although the review is focused on the use of OVs, the general description of OVs is insufficient and  the mechanisms of viral replicative selectivity in neoplastic cells are not well discussed. The antitumoral mechanisms of action of OVs, also, need to be better described. In particular, the capability of OVs to elicit an immune response and to modulate the tumour microenviroment clearly deserves a better discussion.

Authors discuss the result obtained in clinical and preclinical studies. I think that authors should focus more on the clinical data and discuss in a separate paragraph the most promising and recent pre clinical results, avoiding to discuss not clinically validated viruses.  Moreover, authors have very briefly discussed the data obtained with DELYTACT, the only OV so far approved for glioblastoma treatment.  Authors must discuss in detail the data obtained with this virus and possible new strategies based with this virus.

Despite the review should be focused on the use of OVs authors have also discussed other non virally, based approaches for the therapy of GBM,  in this case authors must focus  more on the possibility to combine OVs and novel therapeutic strategies,  alternatively these sections should be eliminated

Minor

Lane 48 and ref 26, A more recent review on hystorical data regarding the capability of viruses reports of  several observation is now available History of how viruses can fight cancer: From the miraculous healings to the approval of oncolytic viruses. Bifulco M, et al Biochimie. 2022 Oct 20:S0300-9084(22)00275-9.

Round 2

Reviewer 2 Report

Authors have reimproved the manuscript. 

I still have a criticisms regarding the sentence on Saint Peregrine Tumour, anecdotal reports in the historical medical literature emphasized that, occasionally, cancer patients showed a clinical remission as a result of natural infections both bacterial or viral. This  phenomenon was called Saint Peregrine tumor and indicates the effects of bacterial and viral infections, Since the manuscript is focused  on the role of oncolytic viruses I suggest to modify the sentences in  line 53-56 with : Several  studies, already at the beginning of  nineteenth centur,y have shown that viral infections can lead to the regression of neoplastic lesions

Author Response

We agree with this reviewer’s notion and have edited the intended sentence to better fit the purpose of the review.